# Survival Outcomes after Hyperthermic Intraperitoneal Chemotherapy for a First Ovarian Cancer Relapse: A Systematic Evidence-Based Review

**DOI:** 10.3390/cancers14010172

**Published:** 2021-12-30

**Authors:** Jean-Marc Classe, Bernard Asselain, Loic Campion, Dominique Berton, Jean-Sébastien Frenel, Fabrice Lécuru, Gwenael Ferron, Laurence Gladieff, Charlotte Bourgin, Cecile Loaec

**Affiliations:** 1Department of Surgery, Institut de Cancerologie de l’Ouest, Boulevard Professor Monod, 44805 Saint Herblain, France; charlotte.bourgin@ico.unicancer.fr (C.B.); cecile.loaec@ico.unicancer.fr (C.L.); 2Faculty of Medicine, Nantes University, 1 Rue Gaston Veil, 44000 Nantes, France; 3Department of Statistics, Arcagy-Gyneco, 75008 Paris, France; b.asselain@gmail.com; 4Department of Statistics, Institut de Cancerologie de l’Ouest, Boulevard Professeur Monod, 44805 Saint Herblain, France; loic.campion@ico.unicancer.fr; 5Department of Medical Oncology, Institut de Cancerologie de l’Ouest, Boulevard Professeur Monod, 44805 Saint Herblain, France; dominique.berton@ico.unicancer.fr (D.B.); jean-sebastien.frenel@ico.unicancer.fr (J.-S.F.); 6Department of Surgery, Institut Curie, Rue d’Ulm, 75005 Paris, France; fabrice.lecuru@curie.fr; 7Université de Paris, 12, Rue de l’École de Médecine, 75006 Paris, France; 8Department of Surgery, Institut Claudius Regaud, Institut Universitaire du Cancer de Toulouse, 31100 Toulouse, France; ferron.gwenael@iuct-oncopole.fr; 9Department of Medical Oncology, Institut Claudius Regaud, Institut Universitaire du Cancer de Toulouse, 31100 Toulouse, France; Gladieff.Laurence@iuct-oncopole.fr

**Keywords:** ovarian cancer, first relapse, hyperthermic intraperitoneal chemotherapy, secondary surgery, progression-free survival, disease-free survival

## Abstract

**Simple Summary:**

Since 2000, scientific literature has recommended the use of hyperthermic intraperitoneal chemotherapy (HIPEC) in the treatment of ovarian cancer relapse. This treatment, combining heavy abdominal surgery and intraperitoneal heated chemotherapy is associated with a risk of post-operative death and severe morbidity. Previous systematic reviews of the scientific literature concluded that HIPEC was effective for improving patient survival following a first relapse of ovarian cancer. This current systematic review, emphasizing the level of evidence of the published series, using the Oxford levels of evidence grading system, has made it possible to analyze the weaknesses of this scientific literature. This literature is characterized by biases—such as patient inclusion, and weak methods—such as retrospective patient collection, a small number of included patients, and no statistical hypothesis. As a results, HIPEC must remain an experimental procedure in ovarian cancer relapse, patients until there are positive results from ongoing clinical trials.

**Abstract:**

Background: Hyperthermic intraperitoneal chemotherapy (HIPEC) is routinely used in the treatment of a first ovarian cancer relapse. Methods: This systematic review, in accordance with Preferred Reporting Items for Systematic Reviews and Meta-analyses guidelines, aimed to assess the quality of scientific proof of the survival benefits of HIPEC, using Medline and Google Scholar. Qualitative analysis using the Oxford CEBM Levels of Evidence 2011 grading is reported. Results: Of 469 articles identified, 23 were included; 15 based on series of patients treated with HIPEC without a control group, and 8 case control series of patients treated with or without HIPEC. The series without a control group showed median overall survival (OS) ranged from 23.5 to 63 months, highlighting a broad standard deviation. Considering the case control series, OS was significantly better in the HIPEC group in 5 studies, and similar in 1. The current review showed considerable heterogeneity and biases, with an Oxford Level of Evidence grading of 4 for 22 selected series and 2 for one. Conclusions: There is no strong evidence to suggest efficacy of HIPEC in improving survival of patients treated for a first relapse of ovarian cancer due to the low quality of the data.

## 1. Introduction

Hyperthermic IntraPeritoneal Chemotherapy (HIPEC) has been used for ovarian cancer relapse since 2000, although it is still not part of standard treatment [1].

Historically, first line standard treatment with complete resection surgery and intravenous bi chemotherapy, +/− bevacizumab, fails to prevent the spread of this peritoneal tumor. Two thirds of patients treated for advanced ovarian cancer experience a relapse, mostly in the form of peritoneal carcinomatosis, suggesting peritoneal failure in standard treatments [2].

HIPEC may improve peritoneal treatment. HIPEC is based on complete surgery followed by high concentration intraperitoneal chemotherapy enhanced with hyperthermic shock, resulting in homogeneous intraperitoneal diffusion [3].

In the context of ovarian cancer relapse, numerous retrospective and prospective studies, leading to reviews and meta-analyses, have bolstered support for HIPEC arguing that there are promising survival results in patients with recurrent ovarian cancer [4,5].

However, the specific role HIPEC plays is difficult to analyze because of a lack of randomized trials and non-standardized methodology, the use of different drugs, concentrations, temperatures, and durations, thus leading to heterogeneous and non-comparable studies [1].

The main objective of the current systematic review was to critically assess the level of scientific proof of survival benefits for HIPEC in the treatment of first ovarian cancer relapse using the Oxford levels of evidence grading system [6].

## 2. Methods

### 2.1. Research Strategy

We conducted a systematic review in accordance with Preferred Reporting Items for Systematic Reviews and Meta-analyses (PRISMA) guidelines. A literature review was carried out using the following key words: ovarian cancer relapse, hyperthermic intraperitoneal chemotherapy, hyperthermia, cytoreductive surgery, survival (overall and progression-free), using Medline and Google Scholar up to December 2020. In accordance with the PICO strategy, our question was defined as follows: population; patients treated for a relapse of ovarian cancer, intervention; treatment must contain HIPEC, comparator; HIPEC characteristics, such as chemotherapy used, posology, temperature and duration, outcomes; survival [7].

### 2.2. Study Selection

One author (JMC) assessed the titles and abstracts to ensure coherence with the inclusion and exclusion criteria, and then two authors (JMC and CL) independently reviewed the full text articles to verify eligibility.

The PRISMA based flow-diagram in Figure 1 shows our research strategy.

Articles not in English, articles without patients, only pharmaceutical results, articles without survival information, articles based on non-epithelial ovarian cancer, or abstracts only were excluded. In cases of mixed populations, the survival results of the group with a first relapse had to be identifiable. In cases of articles from the same institution, the most recent was included to avoid redundancy. Articles were also identified from the reference section of selected articles.

### 2.3. Data Extraction and Analysis

The parameters studied were: number of patients included, period of patient inclusion, pathological criteria, HIPEC method and duration, intraperitoneal chemotherapy used, concentration, residual tumor, disease-free interval (DFI), overall survival (OS), and progression-free survival (PFS). We plotted the PFS and OS data from trials when both sets of data were available, to calculate the correlation between PFS and OS in studies without a control group. Spearman’s rank correlation coefficient was used for testing the significance of this correlation.

### 2.4. Quality Assessment

In this review, we used the Oxford CEBM Levels of Evidence 2011 grading system to define the quality of the publications selected (Level 1 to 4) [6]. Levels of evidence for assessing treatment benefit are defined as follows: level 1: systematic review of randomized trials (meta-analysis), level 2: individual randomized trials, level 3: non-randomized controlled cohort, level 4: case series, case-control studies.

## 3. Results

### 3.1. Series Characteristics

A total of 642 results were identified. After excluding papers with no survival results in the title, 423 papers were screened. Based on the abstract, looking for our inclusion and exclusion criteria, 36 articles were assessed for eligibility. Of the 36 articles, thirteen were excluded for the following reasons:Piso 2004 [8]: In this series, 11 patients had additional courses of intraperitoneal chemotherapy after HIPEC.Raspagliesi 2006 [9]: A series of 40 patients treated for advanced ovarian cancer (*n* = 13) or a relapse (*n* = 27). Survival results could not be identified separately.Cotte 2007 [10]: A series of 81 patients with a mix of first, second or additional relapsed ovarian cancer, and 8 patients with a first relapse not identified separately.Di Georgio 2008 [11]: A series of patients possibly merged with Di Georgio 2017Pavlov 2009 [12]: A series of 56 patients with a mix of first, second, or additional relapsed ovarian cancer, and 25 patients with a first relapse not identified separately.Roviello 2010 [13]: A series of 53 patients treated for advanced ovarian cancer (*n* = 45) or a relapse (*n* = 8) not identified separately.Carrabin 2010 [14]: A series of 22 patients treated with CRS and HIPEC, 12 for primary ovarian cancer and 10 for relapse, with OS results not identified separately.Fagotti 2011 [15]: A series of patients possibly merged with Petrillo 2016.Frenel 2011 [16]: A series of patients possibly merged with Classe 2015.Ansaloni 2012 [17]: A series of patients with initial treatment or first relapse not identified separately.Warschkow 2012 [18]: A series of patients where the survival of patients with a first relapse, with or without HIPEC, was not identified separately.Gouy 2013 [19]: A series of patients based on granulosa ovarian tumors.Cascales-Campos 2014 [20]: A series of 91 patients treated for advanced ovarian cancer, initially or at the time of the first late relapse, not identified separately.

Finally, 23 articles were included in the final analysis (Figure 1: Flowchart).

### 3.2. Clinical Series without a Group Control

Of the 23 series selected, 15 had no control group, resulting in a total of 912 patients. Considering the pathological information provided in the 15 articles selected: there were no inclusion or exclusion pathological criteria in 5 [8,21,22,23,24], one inclusion criterion (epithelial ovarian cancer (EOC), with no exclusion pathological criteria in 6 [25,26,27,28,29,30], definition of inclusion criteria as EOC and exclusion criteria as non EOC in 1 [31], definition of inclusion criteria as EOC and exclusion criteria as borderline in 2 [32,33], and inclusion criteria only (serous ovarian cancer) in 1 [34].

The 15 articles selected demonstrate an important heterogeneity in the main technical aspects of HIPEC: open or closed abdomen techniques, 7 different chemotherapy concentrations, 4 different average temperatures, and exposure from 30 to 120 min (Table 1). CDDP was used in 12 series at different concentrations, from 50 to 250 mg/m², and for different durations, from 60 to 90 min (Table 1). For a total number of 912 patients, 2 articles included more than 100 patients (*n* = 314 [30], *n* = 179 [24]) and the other articles included a mean of 32.2 patients, with six including less than 20 patients.

Complete cytoreduction, CC0 ranged from 65% to 88.6% with 3 articles that did not include this information (Table 2). The mean disease-free interval (DFI) ranged from 3 to 24 months, with 7 articles that did not include this information (Table 2). Only three series distinguished between early relapse (DFI less than 6 months) and late relapse (DFI more than 6 months) (Table 2).

Median OS ranged from 23.5 to 63 months, and median PFS ranged from 10.8 to 27 months, a factor of more than 2.5 between the lower and higher estimates, both for PFS and for OS. Figure 2 shows the correlation between PFS and OS for the 12 articles where both sets of data were available with a broad data range (Figure 2, Table 2). The article with the best OS showed the best PFS. The correlation between PFS and OS is statistically significant (R = 0.71, *p* = 0.009) (Figure 2).

### 3.3. Clinical Series with Control Groups

Eight articles with a control group were selected, including a total of 234 patients treated with HIPEC compared to 295 patients without HIPEC. Regarding the pathological information provided in the 8 selected articles: there were no inclusion or exclusion pathological criteria in 1 series [35], definition of one inclusion criterion (EOC), with no exclusion pathological criteria in 4 [36,37,38,39], definition of one inclusion criterion (serous ovarian cancer) and one exclusion criterion (non-serous) in 1 [40], exclusion of low grade and borderline tumors in 1 [41] and one series provided BRCA mutation status [42].

Heterogeneity in the main technical aspects of HIPEC was observed in the 8 selected articles: differences included open or closed abdomen techniques, 6 different chemotherapy concentrations, 5 different average temperatures ranging from 41 to 43 °C, and exposure from 30 to 90 min (Table 3). For a total number of 174 patients treated with HIPEC in the 7 non-randomized studies, the mean number of patients included was 24.8, compared to 235 control patients (mean =33.6 patients). Four articles specified the term “consecutive” cohort [18,35,37,42].

Complete cytoreduction CC0 ranged between 58% to 100% of cases (Table 4). The median DFI ranged from 18 to 24 months. OS was reported as a percentage or in months, making it difficult to compare between series (Table 4). PFS was missing in 3 articles. PFS ranged from 15 to 26 months in an article with PFS assessment in the HIPEC group (Table 4). In 2 studies, some patients from the control group were treated only with second line chemotherapy without any surgery [37,42].

## 4. Discussion

### 4.1. HIPEC Is Not Standardized

Interestingly, the open or closed abdomen HIPEC technique was not directly compared in any of the series, and a number of them indiscriminately used either of the two techniques [24,27,30,32]. In the randomized trial in our review, 40 patients underwent HIPEC with the open technique and 20 with the closed [39].

Drugs used in HIPEC had to have been tested to assess the intraperitoneal maximal tolerated dose (MTD). Phase I studies assessing the MTD in HIPEC, were published for cisplatin, carboplatin, and pegylated liposomal doxorubicin. For example, the MTD of cisplatin used in HIPEC ranged from 70 mg/m^2^ to 100 mg/m^2^ based on prospective phase I studies [19,33]. The cisplatin dose used in our review ranged from 50 to 150 mg/m^2^. While there is no published prospective phase I study assessing the MTD of taxane in HIPEC, taxane was used in 4 series in our review [26,35,36,39]. Furthermore, whilst oxaliplatin and mitomycin were used in several of the series we reviewed (Table 1 and Table 3), neither of these drugs have been assessed by phase I studies.

Finally, intraperitoneal drug exposure duration varied in relation to the drug used. For example, cisplatin was used at 100 mg/m^2^ for 60 or 90 min [25,28], at 75 mg/m^2^ for 90 or 60 min [8,24,29], and at 50 mg/m^2^ for 90 or 60 min [22,34] (Table 1).

### 4.2. Weaknesses in the Published Series

Statistical hypotheses, power, or the number of patients required to draw conclusions were never specified in the series published. In the randomized trial, the total number of 120 patients required was not argued, therefore limiting the strength of the evidence [39].

Overall, the series published had small sample sizes (Table 1, Table 2 and Table 3). Only two series included more than 100 patients treated with HIPEC [24,30]. In addition, the series of patients studied were rarely consecutive; a long period was required to include few patients, which presupposes bias in patient selection. For example, Robella et al. included 25 patients over a 16-year period, an average of 1.5 patients per year [23].

The series published were predominantly retrospective, and the main bias was patient selection. Initial treatment of primary ovarian cancer, in terms of drug, posology, and number of adjuvant or neaodjuvant cycles of chemotherapy, is another reason for heterogeneity between the series. Most series published included a heterogeneous HIPEC indication, with a mix of initial treatment and relapse, and a mix of pathological subtypes (Table 1, Table 2 and Table 3). In the current review, 6 series did not identify any pathological definition for selecting patients other than “ovarian cancer” [8,21,22,23,24]. PFS between the end of the initial treatment and the first relapse, and residual disease after secondary surgery, both significant prognostic factors for ovarian cancer relapse [2], were missing in 9 and 6 of the 23 series selected respectively (Table 2, Table 3 and Table 4).

One of the main limitations in the scientific literature on assessing HIPEC is the lack of a control group, making it impossible to compare patients treated with or without HIPEC (Table 1 and Table 2). To facilitate this comparison, the authors performed case control studies. In the 7 series in the current review with a control group, the number of patients treated with surgery and HIPEC ranged from 14 to 32 patients, compared to 12 to 84 patients treated without HIPEC (Table 3).

Cianci et al., performed a meta-analysis of 6 case control studies and one randomized trial. The 6 case control series were mainly retrospective, with a small sample, multiple chemotherapy regimen used for the HIPEC, a large period of patients inclusion (from 4 years to 14 years), and in two series, no surgery in the control group [4]. The weaknesses of the randomized trial were previously described above. The main criticism of this trial were no statistical hypothesis, no justification of the number of patients required, different drug regimen used during HIPEC.

In our review, median PFS ranged from 10.8 to 27 months, and median OS ranged from 23 to 63 months for patients treated with HIPEC (Table 2, Table 3 and Table 4). We showed a statistically significant correlation between PFS and OS (R = 0.71, *p* = 0.009), but with a large confidence interval, suggesting strong heterogeneity in the inclusion criteria, HIPEC technique, and follow-up procedures. This also suggests that OS is a more reliable and objective endpoint than PFS (Figure 2). A meta-analysis of 17 randomized trials did not establish PFS as a surrogate end point for OS in first line treatment of advanced ovarian cancer [43]. A systematic review of 22 trials, including recurrence, recommended using OS as primary endpoint for clinical trials in advanced or recurrent epithelial ovarian cancer [44].

### 4.3. Methodology of the Current Review

We chose to perform a systematic literature review. A meta-analysis of the series studied would have led to non-reliable or biased conclusions due to heterogeneity in the patient population, small sample sizes, heterogeneity in the main technical aspects of HIPEC, and a lack of well-built randomized trials [45].

Different tools have been developed to assess the quality of single arm studies or non-randomized clinical trials, characterizing not only existing evidence of bias, but also the content of quality assessment tools and the ways that study quality has been assessed and addressed. A simple comprehensive tool is the Oxford Levels of Evidence grading system which aims to help clinicians conduct their own rapid appraisal for systematic reviews [6]. In the current review, we used this tool to define the quality of the series selected. Using the Oxford CEBM level of evidence, each selected HIPEC series was graded level 4, except the only randomized trial, mentioned above, which was graded level 2.

### 4.4. Prospective Randomized Trials

Prospective randomized trials represent the highest level of evidence [6]. Currently, 3 randomized trials have recently closed for inclusion. The HORSE trial in Italy was initiated to demonstrate the benefits of HIPEC on PFS, with 158 patients required (Clinical trials.gov: NCT01539785). The CHIPOR trial in France, with a total of 404 patients required, was initiated to demonstrate the benefits of HIPEC on OS (Clinical trials.gov: NCT01376752). The last trial from the Memorial Sloan Kettering Cancer Center, USA, was initiated to demonstrate the benefits of HIPEC on PFS, with 99 patients required (Clinical trials.gov: NCT01767675).

## 5. Conclusions

Changes in clinical practices must be supported by strong scientific proof in order to safely improve patient treatment choices without excluding other innovative treatments. The current systematic review shows that in the context of ovarian cancer relapse, the series published have demonstrated that HIPEC is not a standardized procedure, with variations in the parameters with a high treatment impact, such as the intraperitoneal drug used, the posology of its drugs, and the temperature and duration of the HIPEC. Due to the poor-quality level of evidence of cohorts or case-control published series, there is no strong scientific evidence to support the clinical benefits of HIPEC on the outcome of patients treated for relapsed ovarian cancer. In the context of ovarian cancer relapse, HIPEC should be performed within the context of clinical research until the publication of results of high level of evidence trials.

## Figures and Tables

**Figure 1 cancers-14-00172-f001:**
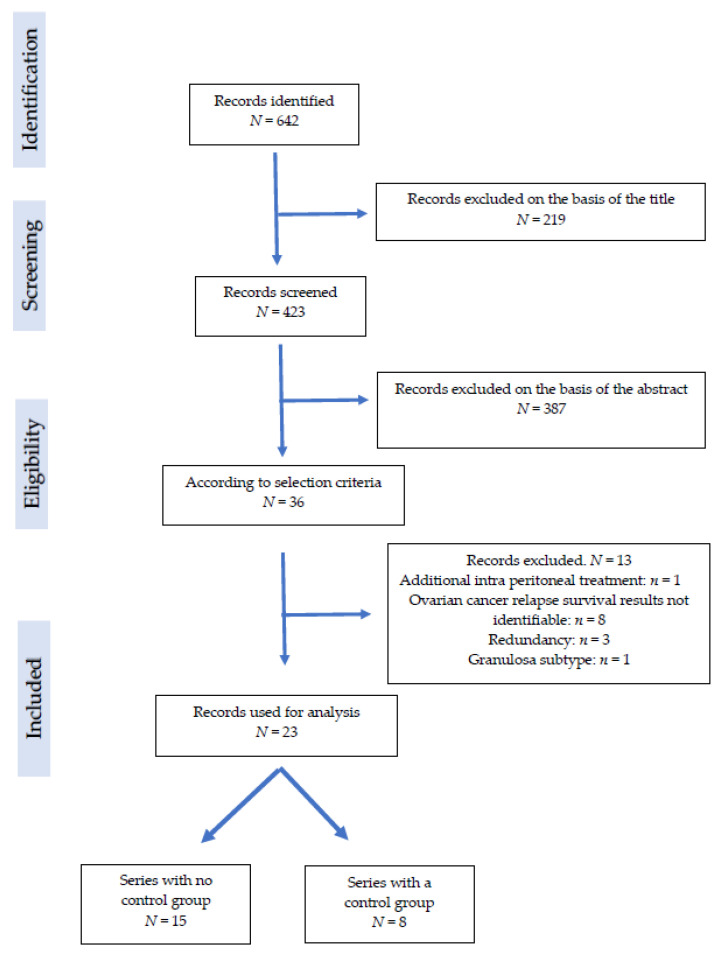
PRISMA flowchart depicting the number of identified articles, those screened and the final number included in the systematic review.

**Figure 2 cancers-14-00172-f002:**
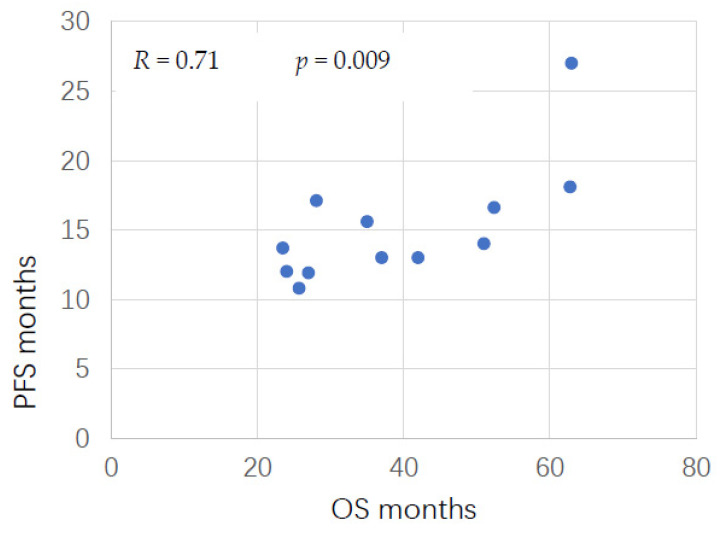
Overall survival (OS) and progression-free survival (PFS) Spearman’s rank correlation in series with no control group. Each dot corresponds to a series in which OS and PFS are available. R is the correlation coefficient.

**Table 1 cancers-14-00172-t001:** Characteristics of series of cytoreductive surgery and hyperthermic intraperitoneal chemotherapy (HIPEC) for ovarian cancer first relapse with no control group. (PIC*O*: Population: ovarian cancer first relapse, Intervention: HIPEC, Comparators: HIPEC characteristics such as intraperitoneal drug, posology, temperature, duration).

Author (Year) (Country)	*n*	OQL	Pro/Retro (Period) M	Systemic Chemo before HIPEC	HIPEC Technique	Drug IP (Poso)	Temp.	Dur. (min)
Zanon (2004) [25] (Italy)	30	4	Pro (January 1998–September 2003)	No	open	CDDP (100 mg/m^2^) (150 mg/m^2^ + Thiosulfate)	41.5 °C	60
Piso (2004) [8] (Germany)	11	4	Retro	Mixed	open	CDDP (75 mg/m^2^) or Mitomycin (15 mg/m^2^)	41.5 °C	90
Rufian (2006) [26] (Spain)	14	4	Retro (January 1997–December 2004)	No	open	Paclitaxel (60 mg/m^2^)	41–43 °C	60
Konigsrainer (2011)12/20/2021 3:31:00 PM (Germany)	31	4	Retro (February 2007–February 2010)	No	open	CDDP (50 mg/m^2^)	42 °C	90
Helm (2010) [32] (Germany)	83	4	(September 2005–June 2008) M	Mixed	Open closed	Platinum Platinum Mito combinationPlatinum Doxo combination(Posology UK)	NP	From 60 to 120
Ceelen (2012) [27] (Belgium)	42	4	Pro (October 2002–January 2009)	Mixed	Open (if Oxali) Closed (if CDDP)	Oxali (460 mg/m^2^) CDDP (100 to 250 mg/m^2^)	41 °C	30 (if Oxali) 90 (if CDDP)
Deraco (2012) [31] (Italy)	56	4	Retro (April 1995–May 2010) M	Mixed	Closed	CDDP + Doxo CDDP + Mito	42.5 °C	90
Gonzalez Bayon (2013) [28] (Spain)	19	4	Retro (June 2002–October 2011)	No	Open	CDDP (100 mg/m^2^) + Doxo (30 mg/m^2^)	42 °C	90
Robella (2014) [23] (Italy)	25	4	Retro (October 1995–December 2011)	No	Closed	CDDP (100 mg/m^2^) + Doxo (15.2 mg/m^2^)	41.5 °C	60
Massari (2014) [21] (Italy)	11	4	Retro (October 2006–December 2009)	Mixed	Closed	CDDP + Caelix Docetaxel + Caelix	42.5 °C	60
Zivanovic (2014) [33] (Germany)	12	4	Pro (October 2011–January 2013)	No	Closed	CDDP (60 mg/m^2^ or 80 mg/m^2^ or 100 mg/m^2^	41–43 °C	90
Classe (2015) [30](France)	314	4	Retro (January 2001–December 2010) M	Mixed	Closed (25%) Open (75%)	NP	42 °C (median)	90 (median)
Delotte (2015) [34] (France)	15	4	Retro (January 2012–January 2014)	No	Open	CDDP (50 mg/m^2^) Doxo (15 mg/m^2^)	43 °C	60
Petrillo (2016) [29](Italy)	70	4	Retro (December 2004–June 2015)	No	NP	CDDP (75 mg/m^2^) Oxali (460 mg/m^2^)	41.5 °C	60 (CDDP) 30 (Oxali)
Di Georgio (2017) [24] (Italy)	179	4	Retro (December 1998–December 2014) M	Mixed	Closed Open Semi closed	CDDP alone (75 mg/m^2^) CDDP mixed with Doxo, Pacli, Mito Oxali (460 mg/m^2^)	UK	60 (CDDP) 30 (Oxali)

*n*: Number of patients studied; OQL: Oxford quality level (1, 2, 3, 4); Pro/Retro: Prospective/retrospective (date of first and last patient inclusion); M: Multicentric. NAC: Neoadjuvant chemotherapy; HIPEC technique: Open or closed abdomen technique (O/C); Drug (Poso): Intraperitoneal chemotherapy (posology, concentration), CDDP (cisplatin), Mito (mitoxantrone), Doxo (doxorubicine), Pacli (paclitaxel), Oxali (oxaliplatine), NP: Not specified; Temp.: HIPEC temperature (°C); Dur.: HIPEC duration (minutes).

**Table 2 cancers-14-00172-t002:** Series of cytoreductive surgery and hyperthermic intraperitoneal chemotherapy (HIPEC) for ovarian cancer first relapse with no control group: residual tumors and survival results. (*PIC*O: Outcomes OS and PFS).

Author (Year)	*n*	Residual	DFI (Median) (mos)	OS (Median) (mos)	PFS (mos)
Zanon (2004) [25]	30	CC0-CC1 77%	UK	28.1	17.1
Piso (2004) [8]	11	UK	18	30	UK
Rufian (2006) [26]	14	UK	UK	57	UK
Konigsrainer (2011)12/20/2021 3:31:00 PM	31	CC0 65%–CC1 25%	24	24	12
Helm (2010) [32]	83	UK	UK	23.5	13.7
Ceelen(2012) [27]	42	No residual 50%	3(median)	37	13
Deraco (2012) [31]	56	CC0 82%–CC1 12%	<6 (23%) >6 (58%)	25.7(whole population)	10.8 (whole population)
Gonzalez Bayon (2013) [28]	19	CC0 73%–CC1 26%	UK	62.8	18.1
Robella (2014) [23]	25	CC0 78.6%–CC1 12.8%	UK	28	NP
Massari (2014) [21]	11	UK	UK	27	11.9
Zivanovic (2014) [33]	12	CC0 58%–CC18%	>6	At 20 mos66.6%	13.6
Classe (2015) [30]	314	CC0 79%–CC1 19%	<6 (53%) >6 (47%)	5142	1413
Delotte (2015) [34]	15	CC0 60%–CC1 40%	UK	35	15.6
Petrillo (2016) [29]	70	CC0 88.6%–CC1 11.4%	19	63	27
Di Georgio (2017) [24]	179	CC0 79.9%	<12 (20%) >12 (80%)	52.4	16.6

*n*: Number of patients studied; CC0-CC1: Cytoreductive completeness 0, 1; UK: Unknown; NI: Not identified. It is not possible to identify results regarding patients with a first relapse; DFI: Disease-free interval, period from the end of initial treatment to the first relapse (months) (UK: Unknown); OS: Overall survival in years or period of OS in %. PFS: Progression-free survival in years or period of PFS in %; NP: Not provided.

**Table 3 cancers-14-00172-t003:** Characteristics of the series of cytoreductive surgery and hyperthermic intraperitoneal chemotherapy (HIPEC) for ovarian cancer first relapse with a control group. (PIC*O*: Population: ovarian cancer first relapse, Intervention: HIPEC, Comparators: HIPEC characteristics, such as intraperitoneal drug, posology, temperature, duration, no HIPEC groups).

Author (Year) (Country)	*n* (H/no H)	OQL	Pro/Retro (Period)M	Systemic Chemo before HIPEC	HIPEC Technique	Chemo IP (poso)	Temp.	Dur. min
Non randomized								
Munoz Casares (2009) [36] (Spain)	14/12	4	Retro (January 1997–December 2004)	No	Open	Pacli (60 mg/m^2^)	41 °C/43 °C	60
Fagotti (2012) [37] (Italy)	30/37	4	Retro (May 2005–October 2009)	No	Closed	Oxali (460 mg/m^2^)	41.5 °C	30
Safra (2014) [42] (Israel)	27/84	4	Retro (UK)	UK	Closed	CDDP (50 mg/m^2^) + Doxo (15 mg/m^2^); Pacli (60 mg/m^2^) + Carboplatinum (AUC4)	42.5 °C	60
Lebrun (2014) [40] (France)	23/19	4	Retro (June 1997–July 2011)	Yes	Open	CDDP Oxaliplatinum	42 °C	6030
Baiocchi (2016) [38] (Brazil)	29/50	4	Retro (May 2000–January 2014)	Mixed	Closed	CDDP (50 mg/m^2^) + Mito (10 mg/m^2^); CDDP (50 mg/m^2^) + Doxo; CDDP alone; Oxali alone	41–42 °C	90
Cascales Campos (2015) [35] (Spain)	32/22	4	Retro (January 2001–July 2012)	Mixed	Open	Pacli (60 mg/m^2^)	42 °C	-
Marocco (2016) [41] (Italy)	19/11	4	Retro (1995–2012)	No	Semi-closed	CDDP (100 mg/m^2^) + Doxo (15 mg/L)	41.5 °C	60
Randomized								
Spilliotis (2014) [39] (Greece)	60/60	2	Pro randomized (2006/2013)	No	Open (*n* = 40) Closed (*n* = 20)	CDDP (100 mg/m^2^) + Pacli (175 mg/m^2^); Doxo (35 mg/m^2^) + Pacli (175 mg/m^2^); Mito (15 mg/m^2^) + Pacli (175 mg/m^2^)	42.5 °C	60

*n*: Number of patients studied (H: In the HIPEC group/No H: In the no HIPEC group), OQL: Oxford quality level (1, 2, 3, 4); Pro/Retro: Prospective/retrospective (date of first and last patient inclusion); M: Multicentric. NAC: Neoadjuvant chemotherapy; UK: Unknown; HIPEC technique: Open or closed abdomen technique (O/C); IP: Intraperitoneal. Drug (poso): Intraperitoneal chemotherapy (posology, concentration), CDDP (cisplatin), Mito (mitoxantrone), Doxo (doxorubicin), Pacli (paclitaxel), Oxali (oxaliplatine); Temp.: HIPEC temperature (°C); Dur.: HIPEC duration (minutes).

**Table 4 cancers-14-00172-t004:** Series of cytoreductive surgery and hyperthermic intraperitoneal chemotherapy (HIPEC) for ovarian cancer first relapse with a control group: residual tumors and survival results. (PICO: Outcomes OS and PFS).

Author (Year)	*n* (H/no H)	Residual	DFI (mos)	OS	PFS (mos)
Not randomized					
Munoz Casares (2009) [36]	14/12	CC0: (H) 64%/(No H) 58%	UK	5 years 58%/17% (*p* = 0.046)	NP
Fagotti (2012) [37]	30/37	CC0: (H) 96.7%/(No H) 100%)	20/22	5 years 68%/42% (*p* = 0.017)	26/15 (*p* = 0.004)
Safra (2014) [42]	27/84	UK	24/21	5 years 79%/45% (*p* = 0.016)	15/6 (*p* = 0.001)
Lebrun (2014) [40]	23/19	CC0: (H) 65%/(No H) 42%	>12 (19/18)	4 years 75.6%/19.4% (*p* = 0.013)	NP
Baiocchi (2016) [38]	29/50	CC0: (H) 79% (No H) 74%	28	58 mos/59 mos (*p* = 0.95)	15.8/18.6 (*p* = 0.82)
Cascales Campos (2015) [35]	32/22	UK	22	NP	3 years 45%23% (NS) (*p* = 0.078)
Marocco (2016) [41]	19/11	CC0: 100%	22/26.9	51.5/NP	19.9/23
Randomized					
Spilliotis (2014) [39]	60/60	CC0: (H) 65%/(No H) 55%	UK	26.7 mos/13.4 mos, (*p* = 0.006)	NP

*n*: Number of patients studied (H: In the HIPEC group/No H: In the no HIPEC group). CC0-CC1: Cytoreductive completeness 0, 1; UK: Unknown; DFI: Disease-free interval, period between the end of the initial treatment and the first relapse (months); OS: Overall survival in years or period of OS in %; PFS: Progression-free survival in years or period of PFS in %; NP: Not provided.

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
