# Peer review of "Survival Outcomes after Hyperthermic Intraperitoneal Chemotherapy for a First Ovarian Cancer Relapse: A Systematic Evidence-Based Review"

_cancers, 2021, doi:10.3390/cancers14010172_

Round 1
Reviewer 1 Report
The authors have addreesed all the concerns. Thew manuscript is now suitable for publication.
Author Response
We sent the corrected manuscript to a native English for english language checking.
Reviewer 2 Report
The problem of HIPEC efficacy in relapsed ovarian cancer is a topic of the extensive debate and is being tested in ongoing prospective clinical trials. Although the topic is not new, authors tried to look at it from a different perspective, and the conclusion differs from that presented in the previous systematic review and meta-analysis (which concluded that HIPEC had promising results for prolongation of OS). To improve the manuscript I propose to discuss the strength and weakness of Cianci et al. meta-analysis in the paragraph 4.3. Authors put there argumentation that meta-analysis is non-reliable due to heterogeneity of data concerning the methods and indications to HIPEC. This should be discussed more extensively, as the differences between Cianci et al. methodology and the present systematic review have influenced the conclusion. What is the main source of biases in meta-analysis? Which papers included could have potentially the greatest impact on this?
Author Response
We had this paragraph:
Cianci et al, performed a meta-analysis of 6 case control studies and one randomized trial.The 6 case control series were mainly retrospective, with a small sample, multiple chemotherapy regimen used for the HIPEC, a large period of patients inclusion (from 4 years to 14 years), and in two series, no surgery in the control group [4]. The weaknesses of the randomized trial were previously described above. The main criticism of this trial were no statistical hypothesis, no justification of the number of patients
required, different drug regimen used during HIPEC.
Reviewer 3 Report
Pag 1 row 45-47
“Due to the poor quality of the level of evidence in the literature”
Please reformulate the sentence to stress more the poor level and quality of the literature.
“There is not scientific evidence to support….” True but the problem is that there is no strong evidence to suggest efficacy due to the quality of the data. In this way, it seems that the technique is rejected.
Pag 2 row
“One reviewer (JMC) assessed the titles and abstracts to ensure coherence with the inclusion and exclusion criteria. Two reviewers (JMC and CL) then independently reviewed the full-text articles to verify eligibility.”
Please, change reviewer with another word because is confusing.
Pag 3
Row 101-103
“We plotted the PFS and OS data from trials when both sets of data were available, to calculate the correlation between PFS and OS in studies 102
without a control group. “
What kind of correlation: spearman or Pearson? Specify
Page 7
Row 181-182
Please check the figure legend.
Page 8
Row 207
"Complete cytoreduction CC0 was performed in 58% to 100% of cases"
What does the sentence mean? In to…or from to?
Page 10
Row 267
“….his also suggests that OS is a more reliable and objective endpoint than PFS (Figure 2)”
It is not clear the clinical significance of this correlation evidence. The OS correlate with PFS but this seems intuitive enough. And why OS is a more reliable and objective endpoint than PFS?
Avoid repeated sentences in the text and reformulate them
Example “shows that HIPEC is not a standardized technique”
Author Response
Pag 1 row 45-47
“Due to the poor quality of the level of evidence in the literature”
Please reformulate the sentence to stress more the poor level and quality of the literature.
“There is not scientific evidence to support….” True but the problem is that there is no strong evidence to suggest efficacy due to the quality of the data. In this way, it seems that the technique is rejected.
We have change Abstract conclusion:
There is no strong evidence to suggest efficacy of HIPEC in improving survival of patients treated for a
first relapse of ovarian cancer due to the low quality of the data.
Pag 2 row
“One reviewer (JMC) assessed the titles and abstracts to ensure coherence with the inclusion and exclusion criteria. Two reviewers (JMC and CL) then independently reviewed the full-text articles to verify eligibility.”
Please, change reviewer with another word because is confusing.
We have change “Reviewer” for “Author”:
One author (JMC) assessed the titles and abstracts to ensure coherence with the inclusion and exclusion criteria. Two authors (JMC and CL) then independently reviewed the full text articles to verify eligibility.
Pag 3
Row 101-103
“We plotted the PFS and OS data from trials when both sets of data were available, to calculate the correlation between PFS and OS in studies 102
without a control group. “
What kind of correlation: spearman or Pearson? Specify
We used Spearman’s rank correlation.
We added this sentence:
Spearman’s rank correlation coefficient was used for testing the significance of this correlation.
Page 7
Row 181-182
Please check the figure legend.
We modified the legend of figure 2.
Figure 2: Overall survival (OS) and progression-free survival (PFS) Spearman’s rank correlation in series with no control group.
Legend: Each dot corresponds to a series in which OS and PFS are available. R is the correlation coefficient.
Page 8
Row 207
"Complete cytoreduction CC0 was performed in 58% to 100% of cases"
What does the sentence mean? In to…or from to?
We modified the sentence:
Complete cytoreduction CC0 ranged between 58% to 100% of cases (Table 4).
Page 10
Row 267
“….his also suggests that OS is a more reliable and objective endpoint than PFS (Figure 2)”
It is not clear the clinical significance of this correlation evidence. The OS correlate with PFS but this seems intuitive enough. And why OS is a more reliable and objective endpoint than PFS?
We had these two sentences and modified references:
A meta-analysis of 17 randomized trials did not establish PFS as a surrogate end point for OS in first line treatment of advanced ovarian cancer [43]. A systematic review of 22 trials, including recurrence, recommended using OS as primary endpoint for clinical trials in advanced or recurrent epithelial ovarian cancer [44].
Avoid repeated sentences in the text and reformulate them
Example “shows that HIPEC is not a standardized technique”
We modified sentence with “HIPEC is not a standardized technique” in order to reduce the number of
repetitions:
3.3 Clinical series with control groups: ….//… Heterogeneity in the main technical aspects of HIPEC was observed in the 8 selected articles
Round 2
Reviewer 2 Report
You have improved your manuscript in the way, that helps the reader to understand your rationale to write it, despite the results of randomized prospective trials is still missing. Well done.
This manuscript is a resubmission of an earlier submission. The following is a list of the peer review reports and author responses from that submission.
Round 1
Reviewer 1 Report
It is an interesting addition to the literature on Hyperthermic Intraperitoneal Chemotherapy (HIPEC) for ovarian cancer relapse treatment. This systematic review focused on the lack of studies of quality that can be found in the literature. The study was designed to find out evidence about how HIPEC was effective in improving the survival of patients with a first relapse of ovarian cancer. However, to be published into IHERPH, authors should deeply review the whole text taking into account certain aspects:
- Authors are advised to avoid the use of taxative expressions such as must, "selected article demostrated", etc.
- In line 60 authors say "In the setting of ovarian cancer relapse, numerous retrospective and prospective studies, reviews and meta-analyses have bolstered support for HIPEC" but it is not supported in any quote. Please provide it.
- The authors do not define a PECO (populations, exposures, comparators, and outcomes) question. It is highly recommended to define this statement for systematic reviews.
- In the section Search strategy, two different lists are given for defining the keywords used. Please clarify.
- In the point 2.4 authors declare that they used the Oxford CEBM Level of Evidence to define the quality of the selected article. It could be highly recommended to introduce a brief summary of it.
- PRISMA workflow is not clear enough. Some aspects should be reviewed (some arrows are missed, records duplicated, articles assessed for eligibility, additional records identify, etc.). On the other hand, the workflow should be included in the Methods.
- The methodology is not well described in the text. In this sense, point 3.2 shows articles excluded and the reason why there are excluded. It is highly recommended to introduce this information on a table.
- Tables 2 and 4 show repeated information in the two first columns. Please merged them.
- Conclusions are a bit weak.
Reviewer 2 Report
The authors describe a literature survey regarding merits of hyperthermic intraperitoneal chemotherapy for ovarian cancer relapse. The study has high relevance but lacks innovation. Methods described are in sufficient detail. I have pointed out a few major concerns below
1) The series of papers that were used were from different regions and taken at different times. Another variation can also be due to geographic location and what standard regimen the ovarian cancer patients were subjected to first line
2) Was HIPEC with a particular chemotherapy drug more beneficial than the rest. The authors should include that information if it is true.
3) More details on Oxford levels of evidence needs to be provided. What statistical analysis it uses etc
Reviewer 3 Report
The problem of HIPEC efficacy in relapsed ovarian cancer is still under debate, therefore the aim of this systemic review is reasonable. However, the manuscript lacks completely any novelty. Accurate systematic reviews and meta-analysis devoted to this topic have been published in 2016 (Hotouras et al.) and recently in 2020 (Cianci et al.)(they are not being cited by the authors - why?). Until the ongoing trials do not finish, the rationale for another one systemic review in the same topic is doubtful.